# Salt Stress Tolerance in *Casuarina glauca*: Insights from the Branchlets Transcriptome

**DOI:** 10.3390/plants11212942

**Published:** 2022-11-01

**Authors:** Isabel Fernandes, Octávio S. Paulo, Isabel Marques, Indrani Sarjkar, Arnab Sen, Inês Graça, Katharina Pawlowski, José C. Ramalho, Ana I. Ribeiro-Barros

**Affiliations:** 1Computational Biology and Population Genomics Group, cE3c–Centre for Ecology, Evolution and Environmental Changes, Faculdade de Ciências, Universidade de Lisboa, 1749-016 Lisboa, Portugal; 2Forest Research Centre (CEF), Associated Laboratory TERRA, Instituto Superior de Agronomia (ISA), Universidade de Lisboa, 1349-017 Lisbon, Portugal; 3Bioinformatics Facility, University of North Bengal, Siliguri 734013, India; 4Department of Ecology, Environment and Plant Sciences, Stockholm University, 106 91 Stockholm, Sweden; 5GeoBioSciences, GeoTechnologies and GeoEngineering (GeoBioTec), Faculdade de Ciências e Tecnologia (FCT), Universidade NOVA de Lisboa (UNL), 2829-516 Monte de Caparica, Portugal

**Keywords:** actinorhizal plants, *Casuarina glauca*, *Frankia*, Illumina RNA-Seq, salt-tolerance

## Abstract

Climate change and the accelerated rate of population growth are imposing a progressive degradation of natural ecosystems worldwide. In this context, the use of pioneer trees represents a powerful approach to reverse the situation. Among others, N_2_-fixing actinorhizal trees constitute important elements of plant communities and have been successfully used in land reclamation at a global scale. In this study, we have analyzed the transcriptome of the photosynthetic organs of *Casuarina glauca* (branchlets) to unravel the molecular mechanisms underlying salt stress tolerance. For that, *C. glauca* plants supplied either with chemical nitrogen (KNO_3_^+^) or nodulated by *Frankia* (NOD^+^) were exposed to a gradient of salt concentrations (200, 400, and 600 mM NaCl) and RNA-Seq was performed. An average of ca. 25 million clean reads was obtained for each group of plants, corresponding to 86,202 unigenes. The patterns of differentially expressed genes (DEGs) clearly separate two groups: (i) control- and 200 mM NaCl-treated plants, and (ii) 400 and 600 mM NaCl-treated plants. Additionally, although the number of total transcripts was relatively high in both plant groups, the percentage of significant DEGs was very low, ranging from 6 (200 mM NaCl/NOD^+^) to 314 (600 mM NaCl/KNO_3_^+^), mostly involving down-regulation. The vast majority of up-regulated genes was related to regulatory processes, reinforcing the hypothesis that some ecotypes of *C. glauca* have a strong stress-responsive system with an extensive set of constitutive defense mechanisms, complemented by a tight mechanism of transcriptional and post-transcriptional regulation. The results suggest that the robustness of the stress response system in *C. glauca* is regulated by a limited number of genes that tightly regulate detoxification and protein/enzyme stability, highlighting the complexity of the molecular interactions leading to salinity tolerance in this species.

## 1. Introduction

Climate change is unequivocally driving major environmental struggles associated with greenhouse gas (GHG) emissions and their impact on global warming [1,2] and sea-level rise [3]. Together with the accelerated rate of population growth, these changes are imposing a substantial loss of biodiversity and arable land [4,5]. Salinization is one of the major global challenges for agricultural systems. Besides the rising seawater levels, prolonged drought events cause secondary soil salinization as a result of an imbalance between water input (irrigation or rainfall) and use (transpiration) [6]. In this context, the use of saline water and halophytes is an alternative strategy to overcome the projected food and environmental crisis [7].

High salinity is strongly associated with ionic and osmotic stresses [8], which disturb a range of key metabolic functions, primarily associated with photosynthesis, respiration, and mineral absorption [9]. During the first phase of stress, the capacity of the plant to take up water is reduced, with a concomitant reduction of the relative water content (RWC) and growth rate. In the second phase, the accumulation of salt ions in the leaves hampers the distribution of photosynthates, and concomitantly plant growth and survival [10]. The extent of salinity impact depends on several aspects such as, the intensity and duration of the stress, leaf and plant age, as well as plant species and ecotype [6,10].

Actinorhizal plants are a group of perennial dicotyledonous angiosperms covering three orders (Fagales, Cucurbitales and Rosales), eight families, and 25 genera [11]. This group is widely distributed worldwide, being able to thrive under extremely harsh conditions, from polar to desert environments. Most of the tree species are pioneer elements in several ecosystems and have been successfully used in land reclamation for a long time [12,13,14,15,16,17]. The environmental resilience of actinorhizal tree species has been associated with their capacity to establish root-nodule symbiosis with N_2_-fixing *Frankia* bacteria, which promotes biomass accumulation and soil fertility [16,18]. Therefore, they constitute a powerful tool to restore degraded lands and at the same time an intriguing model of evolutionary adaptation.

Within the actinorhizal group, *Casuarina glauca* Sieb. ex Spreng. (family Casuarinaceae), a fast-growing tropical tree originating in the southeastern coastal regions of Australia, is considered the model species [19]. Its small genome and the availability of a genetic transformation system make *C. glauca* well suited for basic research. In addition, the biology of its root-nodule symbiosis with *Frankia* has been studied extensively [20]. Moreover, *C. glauca* has an outstanding capacity to endure extreme environments, particularly saline conditions, as it tolerates seawater levels of salt and is commonly planted in areas with a shallow, saline water table [12,19,21,22,23]. Therefore, during the last decade our lab has been dedicated to the elucidation of the mechanisms underlying salt tolerance in this species, as well as the extent of the contribution of symbiotic *Frankia*. Such studies revealed that *C. glauca* withstands up to 400 mM NaCl with remarkably low tissue dehydration, associated with significant osmotic adjustments that minimize the impact of salt on cell water content and the photosynthetic machinery [22]. Besides that, salt tolerance is closely related to the maintenance of cellular membranes’ stability and to a robust anti-oxidative response owing to the accumulation of ROS scavenging components [24], osmolytes, such as neutral sugars, proline, and ornithine [25,26], and flavonoid-based metabolites, i.e., flavonols and proanthocyanidins [27]. Moreover, depending on the plant and bacterial genotype, the ability of *C. glauca* to endure high NaCl concentrations might be coupled or not to the presence of symbiotic *Frankia* [7,28,29,30].

The availability of high-throughput techniques, with a high level of precision and at increasingly affordable prices is currently one of the most straightforward strategies to understand molecular changes of plants in response to salinity at the system level. Among them, RNA-Seq is the main choice for studying phenotypic variations, through the detection of differentially expressed genes (DEGs) between specific environmental conditions [31]. Additionally, a comprehensive annotation of the genome allows the analysis of the functions and activities of all types of transcripts [32], offering a solid basis for the prediction of biomolecular functions and the reconstruction of metabolic pathways [33,34]. Recent transcriptomic studies provided relevant information about the mechanisms underlying the ability of halophyte plants to endure high salt concentrations. Although gene interactions are complex and vary according to plant organ, age, and species, key genes related to salt-tolerance are often related to hormone signaling pathways that regulate cell homeostasis and prevent oxidative damage [35,36,37].

In the present work, we have analyzed the transcriptome of the photosynthetic organs of *C. glauca* (branchlets), one of the first stress targets, to unravel the molecular mechanisms underlying stress tolerance, in complement to previous studies in this species, thus contributing to halophyte research. For that, *C. glauca* plants either supplied with chemical nitrogen (KNO_3_^+^) or nodulated by *Frankia* (NOD^+^) were exposed to a gradient of salt concentrations (200, 400, and 600 mM NaCl) and RNA-Seq was performed.

## 2. Results

### 2.1. Overall Transcriptome Profiling and De Novo Assembly

Quality assessment, data filtering and trimming generated between 22.9 and 26.6 million clean reads (from 33.4 and 37.0 million raw reads) in NOD^+^ plants and between 24.5 and 27.4 million clean reads (from 33.5 to 38.7 million raw reads) in KNO_3_^+^ plants (Appendix A). Minimum per base sequence quality was improved from 2–28 in raw reads to 31–33 in clean reads. No reads were flagged as poor-quality reads. Since FastQ Screen showed no relevant contaminants, all trimmed reads were used in the assembly. De novo assembly showed 41% GC content within a total of 181,484 contigs, 86,202 genes and a contig N50 size of 2792 bp (Appendix A). More than 96% of the reads were mapped back to the transcriptome (Appendix A) with almost 95% completeness (Appendix A).

### 2.2. Differential Gene Expression in Response to Increasing NaCl Concentrations

Principal Component Analyses (PCA) showed a clear separation of samples in four different quadrants: (i) control (0 mM NaCl) and 200 mM NaCl KNO_3_^+^ plants (thereafter referred as 200- KNO_3_^+^); (ii) control and 200 mM NaCl NOD^+^ plants (thereafter referred as 200- NOD^+^); (iii) 400 and 600 mM NaCl KNO_3_^+^ plants (thereafter referred as 400- and 600- KNO_3_^+^); and (iv) 400 and 600 mM NaCl NOD^+^ plants (thereafter referred as 400- and 600- NOD^+^). PC1 accounted for 86% of the total variance, with a clear distinction between the control and 200 mM NaCl samples from the 400 and 600 mM NaCl samples. PC2 (9% variance) distinguished KNO_3_^+^ from NOD^+^ plants (Appendix A).

Overall, the number of Differentially Expressed Genes (DEGs) increased with increasing salinity (Figure 1A). All DEGs (FDR < 0.05) showed a log2FC lower than −2 or higher than 2 (Appendix A). KNO_3_^+^ plants expressed a total of 19,913 genes in the control and an average of 19,645 genes in samples exposed to salinity (Table 1). From these, an average of 15,328 genes was expressed in both control and salinity-exposed plants. The percentage of significant DEGs was extremely low, i.e., 0.04% (9), 1% (238) and 2% (359) in 200-, 400- and 600- KNO_3_^+^, respectively. All DEGs were downregulated at 200 mM NaCl. At 400 mM NaCl and 600 mM NaCl, ca. 88% (210 and 314, respectively) DEGs were downregulated and ca. 13% (28 and 45, respectively) upregulated (Figure 1A and Table 1).

NOD^+^ plants expressed a higher number of genes compared to KNO_3_^+^ plants, with a total of 20,278 genes at 0 mM NaCl and an average of 19,780 genes in samples exposed to salinity (Table 1). From those, an average of 16,023 genes was common to control and salinity-stressed plants. Similar to KNO_3_^+^ plants, a decreasing number of common genes and an increasing number of DEGs were observed with increasing salinity. The percentage of significant DEGs at all salinity conditions was slightly lower than in KNO_3_^+^ plants: 0.03% (6), 0.5% (104), and 1.3% (254), respectively in 200, 400, and 600 mMNOD^+^. At 200 mM NaCl, the number of up-and down-regulated DEGs was similar. At 400 and 600 mM NaCl, the percentage of downregulated vs. upregulated DEGs was 60% (62) vs. 40% (42), and 85% (217) vs. 15% (37), respectively.

The number of DEGs specific to only one salt concentration (200, 400, or 600 mM NaCl), increased with the salinity in both plant groups (Figure 1B). In KNO_3_^+^ plants, the number of treatment-specific downregulated DEGs was 1 (0.3%), 69 (18%), and 175 (45%) at 200, 400, and 600 mM NaCl, respectively. Additionally, two (0.5%) DEGs were commonly found in 200- and 400 KNO_3_^+^, six (2%) in all set of conditions, and 133 (35%) in 400- and 600 KNO_3_^+^. Together, the two highest salt concentrations accounted for ca. 98% (175 + 133 + 69) of all downregulated DEGs. Among the upregulated DEGs, 11 (18%) were activated in both 400- and 600- KNO_3_^+^, 17 (27%) were specific to 400 mM NaCl and 34 (55%) were specific to 600 mM NaCl (Figure 2B). In NOD^+^ plants, treatment-specific downregulated DEGs increased with the salinity level: three (200 mM), 62 (400 mM), and 218 (600 mM). Among these, 27 (11%) were specifically found in 400 NOD^+^, and 183 (75%) in 600 NOD^+^ (Figure 2C). Specific downregulated DEGs were not found in 200-NOD^+^. Only 1 (0.4%) DEG was commonly found in 200- and 400 NOD^+^, 2 (0.8%) in all sets of conditions, and 32 (13%) in 400- and 600 NOD^+^. Together, the number of specific DEGs in 400- and 600- NOD^+^ accounted for 99% (27 + 32 + 183) of the downregulated pool. Among the upregulated DEGs, only 1 (1.5%) was found in 200 NOD^+^, 30 (44%) in 400 NOD^+^, and 25 (37%) in 600 NOD^+^ (Figure 2D). The transcriptional activity of two (3%) DEGs was activated in samples from all set of conditions and 10 (15%) in 400- and 60 NOD^+^. No common DEGs were identified between the other set of conditions. Together, 400- and 600 mM NOD^+^ accounted for 96% (25 + 10 + 30) of the treatment-specific DEGs.

### 2.3. Differential Gene Expression between KNO_3_^+^and NOD^+^ Plants

Comparing KNO_3_^+^ and NOD^+^ plants, no common DEGs were observed in samples exposed to 200 mM NaCl (Figure 3B), while 32 (10%) and 135 (28%) were observed in samples subjected to 400 and 600 mM NaCl, respectively (Figure 3B,C). The number of specific DEGs was higher in KNO_3_^+^ than in NOD^+^: nine vs. six, 206 (67%) vs. 72 (23%), and 224 (47%) vs. 119 (25%), at 200, 400, and 600 mM NaCl, respectively (Figure 3, Appendix A).

### 2.4. Top NaCl-Responsive Genes in KNO_3_^+^ and NOD^+^ Plants

In KNO_3_^+^ plants, only 46% (208 out of 448) DEGs were uniquely mapped to UniProtKB/Swiss-Prot database proteins, two of which corresponded to yet uncharacterized proteins (Appendix A). Considering each salt condition, 2 (22%), 125 (53%) and 164 (46%) mapped DEGs were identified in samples exposed to 200, 400, and 600 mM NaCl, respectively (Table 1). In 200 KNO_3_^+^, the most downregulated DEGs were ERF020 (log2FC: −4.06), related to stress signal transduction pathways, and GT-3B (log2FC: −3.81), a salt responsive transcription factor. In 400 KNO_3_^+^ the top responsive DEGs were WAXY (log2FC: 3.46) and XCP1 (log2FC: −6.88), respectively. The first is associated with starch biosynthesis, while the second is related to programmed cell death and proteolysis. In 600- KNO_3_^+^, *XCP1* (log2FC: −6.88) was also the most downregulated DEG, and PLT6, which is related to glucose import, the most upregulated one (log2FC: 4.55). The top 10 most down- or upregulated DEGs for KNO_3_^+^ plants are summarized in Table 2 (400 mM NaCl) and Table 3 (600 mM NaCl), while the full list of DEGs can be found in Appendix A.

Heatmaps were used to analyze the regulation pattern of the top 10 salt-responsive DEGs from 600 mM NaCl relative to the control. For each treatment, the normalized read counts of these genes were also plotted to allow the comparison of gene regulation over the salinity gradient (Appendix A). Two main clusters were formed, one with samples from 0 mM and 200 mM NaCl, and another with samples from 400 mM NaCl and 600 mM NaCl. The top downregulated DEGs from KNO_3_^+^ plants exposed to 600 mM NaCl included an association with response to hypoxia (GO:0001666), DNA (GO:0003677), carbohydrate metabolism (GO:0030246), metal ion- (GO:0046872) and mannose binding (GO:0005537), leaf senescense (GO:0010150), cell death (GO:0008219), cell wall organization biogenesis (GO:0071554), and iron homeostasis (GO:0055072) and -transport (GO:0006810). The top upregulated DEGs were related to the biosynthesis of (−)-secologanin, the precursor of monoterpene indole alkaloids (GO:1900994), abscisic acid-activated signaling (GO:00097389), alkaloid metabolism (GO:0009820), plant-type hypersensitive response (GO:0009626), regulation of stomatal opening (GO:1902456), glucose import (GO:0046323), UDP-glycosyltransferase (GO:0008194), antiporter activities (GO:0015297), chitin catabolism (GO:0006032), and chaperone cofactor-dependent protein refolding (GO:0051085) (Table 3; Appendix A).

In NOD^+^ plants, 43% (136 out of 313) DEGs were uniquely mapped to known proteins, four of which uncharacterized (Appendix A). Considering each salt condition, four (67%), 49 (47%) and 111 (44%) of mapped DEGs were identified in 200-, 400-, and 600- NOD^+^, respectively (Table 1). Among the annotated DEGs from 200- NOD^+^, the most upregulated gene was PEPR1 (log2FC: 3.59) which is involved in PAMP-triggered immunity (PTI) signaling, followed by GDH2 (log2FC: 3.32) that catalyzes the production of GABA, and CYP78A5 (log2FC: 2.89) required for the promotion of leaf and floral growth and the prolongation of the plastochron. The only downregulated gene was GLPR2.8 (log2FC: −3.25), involved in light signal transduction and calcium homeostasis via the regulation of calcium influx into cells. In 400 NOD^+^, the most responsive DEGs were GDH2 (log2FC: 5.09) encoding a glutamate dehydrogenase, and RL5 (log2FC: −7.79) related to DNA binding. In 600 NOD^+^, the most up- or downregulated DEGs were PEPR1 (log2FC: 4.59) encoding a Leucine-rich repeat receptor-like kinase involved in PTI signaling, and RL5 (log2FC: −7.79), respectively. The top 10 most responsive DEGs in the NOD^+^ group are summarized in Table 4 and Table 5 and the full list of DEGs can be found in Appendix A.

Regarding the top 10 up- or downregulated DEGs, two main clusters were also formed in NOD^+^, one including the 0 mM and the 200 mM NaCl groups, and the other including the 400 mM and 600 mM NaCl groups. However, unlike the situation in KNO_3_^+^ plants, a clear separation of control and 200- NOD^+^ was observed in the first cluster. Similar to KNO_3_^+^ plants, the top downregulated DEGs in NOD^+^ plants exposed to 600 mM NaCl were associated with response to hypoxia, DNA- and mannose binding, cell death and cell wall biogenesis and organization (Table 3 vs. Table 5; Appendix A vs. Appendix A). Additionally, these DEGs were also related to defense responses (GO:0006952), leaf morphogenesis (GO:0009965), response to oxidative stress (GO:0006979), cellular response to amino acid stimulus (GO:0071230), phosphorylation (GO:0016310), defense responses to bacteria (GO:0042742), signal transduction (GO:0007165) and metabolic processes (GO:0008152) (Table 5; Appendix A). Also comparable to KNO_3_^+^ plants, in NOD^+^ plants the top upregulated DEGs at 600 mM NaCl were related to abscisic acid-activated signaling (GO:0009738), alkaloid metabolism (GO:0009820) and regulation of stomatal opening (GO:1902456) (Table 3 vs. Table 5; Appendix A vs. Appendix A). Additionally, they were also associated with response to sulphur starvation (GO:0010438), jasmonic acid (GO:0009753) and wounding (GO:0009611), protein transport (GO:0015031), chaperone cofactor-dependent protein refolding, transmembrane transporter activity (GO:0022857), carbohydrate metabolism (GO:0005975), cell growth (GO:0016049), and lumen of the endoplasmic reticulum (GO:0005788) (Table 5; Appendix A).

Only two out of the top 10 most downregulated DEGs in samples exposed to 400 mM NaCl were found in both plant groups, namely *RL5* and *FLA12*, both related to plant-type secondary cell wall biogenesis. Among the top upregulated DEGs, only the sulphur deficiency induced gene *SDI1* was common to both groups. At 600 mM NaCl, half of the top downregulated DEGs were common to both KNO_3_^+^ and NOD^+^, namely *RL5*, Lectin, *LAC22*, *LECRK91*, and *LRK10L-1.4*. These genes were respectively involved in the following biological processes: DNA binding, iron ion homeostasis and transport, lignin catabolic process, defense response to bacterium and oomycetes, positive regulation of cell death, positive regulation of hydrogen peroxide metabolic process, cellular response to hypoxia and phosphorylation. Regarding the top upregulated DEGs, only two were common to KNO_3_^+^ and NOD^+^ plants, namely DTX56, associated with the cellular response to carbon dioxide and regulation of stomatal opening, and HSP70, involved in cellular response to unfolded protein and chaperone cofactor-dependent protein refolding. Overall, differential expression values, estimated by log2FC, were always higher in downregulated than in upregulated DEGs in both plant series.

### 2.5. Differentially Top-Expressed Genes in KNO_3_^+^ and in NOD^+^ Plants

In KNO_3_^+^ plants, Clust tool showed 10 different clusters (Figure 4, Appendix A). Clusters C0 to C3, C8, and C9 consistently presented a clear downregulation of DEGs at 600 mM NaCl relative to the control, with changes in the regulation pattern at 200- and 400 mM NaCl. Cluster C0 showed a progressive downregulation with increasing salinity from 200 to 600 mM NaCl, while C1 displayed an accentuated downregulation only after 400 mM NaCl. Although clusters C2 and C3 presented a sharp downregulation between 400 and 600 mM, they also showed an increasing upregulation of DEGs from control to 400 mM NaCl, which is much more evident in C3. The downregulation pattern in clusters C8 and C9 was sharp until 400 mM NaCl, nearly maintaining the expression levels at the highest salinity condition. In contrast, clusters C4 and C5 consistently presented a strong upregulation of DEGs from 0 to 600 mM NaCl. Overall, in clusters C6 and C7, a downregulation was observed at 400 mM NaCl, more accentuated in C7, and reverted at 600 mM NaCl in both cases.

DEGs in almost all clusters were associated with DNA-, metal-, or sugar-binding, catalytic activity, transport, signaling, defense against biotic and abiotic stress and response to stimuli. In cluster C0, they were also related to chloroplast components, cell death, aging, oxidation-reduction process, and response to osmotic stress and water deprivation. Cluster C1 was more associated with membrane and mitochondrial components, and response to cold and salt stress. Although presenting similar expression patterns, while C2 was mainly associated with membrane components and defense responses, C3 was related to stomatal movement and signaling pathways involving brassinosteroids, ethylene and abscisic acid. Cluster C4 was associated with membrane and chloroplastic components, stomatal opening, glucose import, response to carbon dioxide, jasmonic acid and auxin, heat acclimation, regulation of growth. However, cluster C5, which presented a similar expression pattern, was mainly related to the response to metabolic processes and oxidative stress. Clusters C6 and C7 were both related to membrane, Golgi apparatus, mitochondria, and chloroplast components. However, while C7 was associated with the cell cycle, response to cold, root hair elongation and cell wall organization, C6 was related to cellular respiration, defense responses and the response to stress, including cold and salt stress. Clusters C8 and C9 were associated with defense responses and membrane, but while C8 was deeply related to the microtubule, auxin homeostasis, and signaling, C9 was more related to cell wall, leaf abscission, and cell death. The full list of GO terms of the three main categories associated with each cluster of DEGs from KNO_3_^+^ plants can be seen in Appendix A.

In NOD^+^ plants, the Clust tool showed 11 different clusters (Figure 5, Appendix A). Clusters C0 to C5 and C10 presented a noticeable downregulation of DEGs at 600 mM NaCl relative to the control, with different patterns of regulation at 200- and 400 mM NaCl. Clusters C0 and C1 showed a behavior similar to that of the corresponding clusters in KNO_3_^+^ plants, where the first had a progressive downregulation with increasing salinity and the second only showed an accentuated downregulation at 600 mM NaCl. Clusters C2 and C3 showed upregulation at 200 mM NaCl, which overall was reverted at higher salinity levels. In clusters, C4 and C5 the major downregulation happened rapidly between 200 and 400 mM NaCl, slightly higher in C4 and moderately reverted in C5 at 600 mM NaCl. In cluster C10, the downregulation occurred in two marked steps, between the control and 200 mM NaCl and between 400 and 600 mM NaCl. Cluster C6 presented a strong downregulation at 400 mM NaCl, which was inverted at the highest salinity concentration to levels close to the control. Cluster C7 showed a somewhat similar behavior. However, the upregulation at 600 mM was much more evident and the transcript levels were higher than those in the control.

Overall, like in the case of KNO_3_^+^ plants, the majority of clusters were associated with binding, catalytic activity, transport, signaling, the defense response and responses to stimuli and stress. Cluster C0 was related to membranes and mitochondria, and highly associated with responses to stimuli, namely salicylic, jasmonic, and abscisic acids, ethylene and chitin, and stress responses, specifically to wounding, hypoxia, water deprivation and salt stress. Cluster C1 included a set of genes associated with chloroplasts, defense response, oxidation-reduction process and cell death. Cluster C2 was mainly associated with the defense response and responses to stimuli and stress, namely cold, hypoxia, wounding, and auxin, also including some genes related to cell differentiation and membrane components. Clusters C3 and C4 were enriched in genes involved in cell death and redox homeostasis. Additionally, C4 was also related to chloroplast, aging, cell wall biosynthesis and organization, response to oxidative stress, wounding, and biotic stimuli. Cluster C5, which was associated with chloroplasts and membrane components, included also genes involved in response to auxin and biotic stimuli, leaf morphogenesis and metabolic processes. Cluster C6 represented DEGs associated with regulation of growth, cell differentiation, and responses to oxidative and salt stress. Cluster C7 was mainly related to cell cycle, protein phosphorylation and the Golgi apparatus. Cluster C8 included genes involved in oxidation-reduction processes, chloroplasts, stomatal opening, and cellular responses to carbon dioxide. Cluster C9, which was related to membrane components, contained also genes related to the response to jasmonic acid and biotic stimuli, DNA protection, and defense against biotic stress. Finally, Cluster C10 included genes associated with membrane components, response to ozone, phloem development, response to hypoxia and abscisic acid. The full list of GO terms of the three main categories associated with each cluster of DEGs from NOD^+^ plants can be seen in Appendix A.

### 2.6. Significantly Enriched GO Terms, Metabolic Pathways and PPIs

Over-representation analysis (ORA) performed in gProfiler for both plants, found a few sets of enriched GO terms including DEGs in samples exposed to 400- and 600 mM NaCl with FDR < 0.05. Due to their low number, no enriched GO terms were found for DEGs in samples exposed to 200 mM NaCl. In 400 KNO_3_^+^ plants, enriched DEGs were related to UDP-glycosyltransferase activity (GO:0008194), cell wall organization (GO:0071555), external encapsulating structure organization (GO:0045229), defense against pathogenic bacteria (GO:0050829), polysaccharide biosynthetic process (GO:0000271), leaf abscission (GO:0060866), cell periphery (GO:0071944), and anchored component of plasma membrane (GO:0046658). In NOD^+^ plants enriched DEGs corresponded to glutamate dehydrogenase (NAD^+^) (GO:0004352), terpenoid catabolic process (GO:0016115), and cell periphery (GO:0005886) (Table 6). In 600- NOD^+^, DEGs from both plant groups were enriched in protein serine/threonine kinase activity (GO:0004674), ATP binding (GO:0005524), purine ribonucleoside triphosphate binding (GO:0035639), adenyl ribonucleotide binding (GO:0032559) and drug binding (GO:0008144), protein phosphorylation (GO:0006468) and defense response, incompatible interaction (GO:0009814). In KNO_3_^+^ plants, DEGs were also enriched in carbohydrate derivative binding (GO:0097367) and abscission (GO:0009838). In NOD^+^ plants, there was an enrichment in all-trans-beta-apo-10’-carotenal cleavage oxygenase transcription (GO:0102251), 9-cis-10’-apo-beta-carotenal cleavage oxygenase activity (GO:0102396), phospholipase transcription (GO:0004620), purine ribonucleotide binding (GO:0032555), as well as processes involving the interaction between organisms (GO:0051704), response to hypoxia (GO:0001666), defense response to bacteria (GO:0042742), positive regulation of defense responses (GO:0031349), lipid catabolic process (GO:0016042), leaf abscission (GO:0060866), and cell death (GO:0008219) (Table 7).

Through ShinyGO, enriched DEGs were observed only at 600 mM NaCl in both plant groups. In both InterPro and Pfam databases, KNO_3_^+^ plants were enriched for the transcription of genes involved in the following pathways: glycosyltransferase family 8, GIY-YIG catalytic domain/nuclease superfamily, UDP-glucoronosyl and UDP-glucosyl transferase, KOW motif, and the unkown function domain DUF2828. Furthermore, a set of Kyoto Encyclopedia of Genes and Genomes (KEGG) pathways was also enriched for this condition, namely cyanoamino acid metabolism, amino sugar and nucleotide sugar metabolism, and starch and sucrose metabolism. NOD^+^ plants were enriched in transcripts of AAA-type ATPase, chalcone/stillbene synthase, flavinoxireductase/NADH oxidase, glutamate/phenylalanine/leucine/valine dehydrogenase, and phosphatase 2C family. All enriched pathways are listed in Appendix A.

Likewise, PPI networks of DEGs were found in the STRING database only at 600 mM NaCl in both plant groups. In 600- KNO_3_^+^ plants, three DEGs associated with enriched biological processes were mapped in the PPI network, namely SAG101, CRT3 and TAO1 (Appendix A). The first two, which were related to the regulation of response to stress and positive regulation of response to stimuli, were directly linked in the network. Moreover, in NOD^+^ plants, these two genes were indirectly linked to RBOHD, which was associated with cell death. Another four DEGs in NOD^+^ plants were mapped in the PPI network, namely SAG101, CRT3, TAO1 and PUB17 (Appendix A). Again, the first two were directly linked in the network. However, in this case, while CRT3 is associated with cell death, SAG101 is related to defense signaling, response to bacteria, regulation of response to stress and positive regulation of response to stimuli. Moreover, PUB17, which was associated with the defense response, was indirectly linked to both of these genes.

### 2.7. Genetic Stability and Volatility among DEGs

CodonW analysis was performed to assess the genetic stability or volatility among DEGs. It was observed that genes upregulated in NOD^+^ plants had a GC content of 69.35% with 58.79% GC content at synonymous sites (GC3). A significant positive correlation between % GC and the frequency of optimal codons (Fop) was found (r = 0.57), whereas a significant negative correlation was obtained between GC3 and the effective number of codons (Nc) (r = −0.62). On the contrary, the upregulated genes in KNO_3_^+^ plants showed different features. In this group, GC was 50.02% with 51.03% GC3. Moreover, statistical analyses between GC and Fop along with GC3 and Nc did note demonstrate statistical significance. This indicated distinct genetic volatility among the upregulated genes present in KNO_3_^+^ plants. Moreover, COG analysis revealed “cellular processing and signaling” as a major functional category followed by “information storage and processing” and “metabolism” in NOD^+^ plants (Figure 6A), whereas “information storage and processing” was found to be the major COG family in non-nodulated KNO_3_^+^ plants (Figure 6B). The upregulated proteins in NOD^+^ plants were more energy-consuming than those in non-nodulated plants.

## 3. Discussion

The impact of salt stress in *Casuarina glauca* has been extensively studied in depth at the morpho-physiological and biochemical levels, as well as at the proteome and metabolome scale [7]. Altogether, these studies revealed the robustness of the stress-responsive mechanisms in this species, particularly regarding protective traits at the photosynthetic and membrane level, whose activation was triggered at early stages of stress. In order to conclude this broad analysis, here, we report on the effects of increasing salt concentrations (200, 400 and 600 mM NaCl) on the transcriptome of branchlets from non-nodulated (KNO_3_^+^) and nodulated (NOD^+^) *C. glauca* plants. RNA-Seq analysis yielded an average of ca. 25 and ca. 26 million clean reads for KNO_3_^+^ and NOD^+^ plants, respectively, corresponding to 86,202 unigenes with a N50 size of 2792 bp and 41% GC content. These values are within the range of those reported in similar studies in *Casuarina*, which vary according to the species, environmental condition, and plant organ. For example, in *Casuarina equisetifolia* seedlings exposed to cold, ca. 21 million reads corresponding to ca. 118,000 unigenes and a N50 value of 2827 bp have been reported [38]. Developing secondary wood tissue of the same species presented ca. 43 million clean reads, comprising ca. 27,000 unigenes with a N50 contig size of 750 bp [39]. On the other hand, in *C. equisetifolia* roots from plants exposed to salt, these values were ca. 75–95 million clean reads comprising ca. 53,000–72,000 unigenes [37]. Finally, Wang et al. [40] reported ca. 41 million clean reads comprising ca. 60,000 unigenes with a N50 value of 2832 bp in arbuscular mycorrhiza roots of *C. glauca* exposed to salt. These results, together with the high mapping coverage of our data (almost 95%) indicates that a high-quality transcriptome assembly has been generated for downstream analyses.

In both plant groups, the patterns of differentially expressed genes (DEGs) clearly separate two groups, (i) control and 200 mM NaCl-treated plants, and (ii) 400 and 600 mM NaCl-treated plants (Figure 4, Figure 5, Appendix A). With a few exceptions, this separation from low-to-moderate impacts (0, 200 mM) and heavy impacts (400, 600 mM) was also reported for several morpho-ecophysiological parameters, like plant growth and water use efficiency, mineral contents, leaf gas exchanges, chlorophyll a fluorescence, thylakoid electron transport rates, photosynthetic enzymes, and structural carbohydrates [22], as well for differentially expressed proteins (DEPs) of the same experimental set [30]. Notably, although the number of total transcripts was relatively high in both plant groups (9930 in the control and an average of 19,645 genes in NaCl-treated KNO_3_^+^ plants, and 20,278 in the control and an average of 19,780 genes in NaCl-treated NOD^+^ plants), the percentage of significant DEGs was remarkably low, ranging from 6 (200 NOD^+^) to 314 (600 KNO_3_^+^), mostly downregulated (Figure 1; Table 1). The results are within the range of those reported for the invasive halophyte, *Phragmites karka*, exposed to 150 mM NaCl: 305 and 289 DEGs in leaves and roots, respectively [41]. Nevertheless, in this case, the fraction of positively regulated genes was higher than that of the negatively regulated ones. On the other hand, the number of DEGs reported in *C. equisetifolia* seedlings was considerably higher: >6000 DEGs in roots from plants exposed to 200 mM NaCl for seven days [37] and >4000 DEGs in leaves of plants exposed to cold [38]. However, in both cases the fold-change, rather than the statistical significance, has been privileged. Such observations highlight the complexity of the molecular interactions leading to salinity tolerance in *C. glauca* which might be related to a rapid and robust stress-responsive system [7], probably linked to long-term ecological adaptation [38].

A minor proportion of DEGs was common to both plant groups, in response to 400 (10%) and 600 mM NaCl (28%) (Figure 3), while no common DEGs were identified at the lowest salt concentration. The fact that the nodulation process triggers a set of defense-responses [42,43] and that at 200 mM NaCl the N2 fixation by *Frankia* is residual [23] may explain this difference. Nevertheless, in both plant groups, the increase of the number of DEGs was gradual along the salt gradient, likely reflecting changes towards acclimation to high salt levels [30].

Considering the two key stress concentrations, i.e., 400 and 600 mM NaCl, the stress-responsive genes were in general related to regulatory processes in both plant groups. The results are rather different from those reported for the *C. glauca* branchlets’ proteome [30], and metabolome [25,26,27], where changes were associated to major physiological changes related to photosynthesis, membrane stability and osmoprotection mechanisms [22,24,25,26,27]. This reinforces the hypothesis that *C. glauca* has a strong stress-responsive system with an extensive set of constitutive defense mechanisms [7,43] complemented by rapid induction of transcriptional changes at the initial stages of salt exposure [37,40]. Despite the differences associated with the molecular changes imposed during the nodulation process (highlighted above), in both plant groups the top-ten up-regulated DEGs were related to signaling, transport, refolding, and stomatal control (Table 1, Table 2, Table 3, Table 4 and Table 5; Appendix A). These included auxin- (400 KNO_3_^+^) and abscisic acid (ABA)-activated signaling (600 KNO_3_^+^; 600 NOD^+^); Myb transcription factors and jasmonic acid signaling (400 NOD^+^), regulation of sulphur utilization, transporter activity, and protein refolding (400- and 600- KNO_3_^+^; 400- and 600 NOD^+^), and response to cellular CO2/regulation of stomatal opening (600 KNO_3_^+^; 600 NOD^+^). It is widely known that the mechanisms used by plants to cope with salt stress are highly dependent on a coordinated set of events that regulates growth, metabolism, and homeostasis [37,44,45]. In halophytic plants, gene regulatory networks are an essential part of salt tolerance. For example, in *C. equisetifolia* [37], *Spartina alterniflora* [46], and *Puccinellia nuttalliana* [47], transcription factors (TFs), and ion transporters were identified as key drivers of salt stress tolerance, playing an essential role in ion homeostasis, regulation of ROS scavenging and detoxification, water balance, and water transport.

The top up-regulated genes common to both plant groups were the sulphur deficiency induced gene, SDI1, the gene encoding protein detoxification 56, DTX56, and the gene encoding the 70 KDa heat shock protein, HSP70 (Table 2, Table 3, Table 4 and Table 5). The accumulation of SDI transcripts is related to sulphur deficiency, utilization of stored sulphate pools and sulphur partioning [48,49,50]. It has already been shown that the expression of the sulfate uptake transporter SULTR3 is induced in *Arabidopsis* and *Medicago* in response to various abiotic stresses [51] and that this induction depends on ABA [52]. Given that levels of cysteine and glutathione tend to increase significantly under abiotic stress conditions, which was also confirmed for glutathione in *C. glauca* [24,30], an upregulation of sulphate use is a plausible response to long-term salinity stress. On the other hand, DTX belongs to the group of transporters involved in detoxification in plants [53,54,55], as well as in turgor-regulation and ABA efflux in drought tolerant Arabidopsis [56]. Thus, it seems reasonable to hypothesize that DTX56 is a key multifunctional gene involved in the modulation of *C. glauca* responses to high salinity through the regulation of signal transduction pathways and associated mechanisms leading to stress tolerance [57]. Finally, the induction of the expression of the chaperone HSP70 is likely related to the maintenance of protein folding/unfolding as well as degradation of misfolded and denatured proteins, a common mechanism observed in plants under stress, e.g., osmotic [58], drought [59], and heat [60].

To be highlighted also is the fact that in NOD^+^ plants, glutamate dehydrogenase (*GDH*) 2 is upregulated under all three salinity levels tested (Table 5 and Table 6, Appendix A). This enzyme is generally involved in stress adaptation [61] and plays an important role in replenishment of the tricarboxylic acid cycle and generally at the interface between carbon and nitrogen metabolism [62]. In fact, these authors showed that during short term salt stress, post-transcriptional regulation of GDH led to increased enzyme activity, and overexpression of the corresponding gene, improving biomass accumulation. Interestingly GDH is upregulated in NOD^+^, but not in KNO_3_^+^ plants.

Abscisic acid 8’-hydroxylase initiates the degradation of ABA and is thus important for maintaining ABA homeostasis [63,64]. The upregulation of its gene in NOD^+^ plants exposed to 400 mM NaCl is interesting as during salt stress, an upregulation of ABA synthesis, not degradation, would have been expected [65], supporting a gradual stomatal closure that reached negligible values at 600 mM NaCl [22]. Another result that might seem surprising is the upregulation of WAXY, the starch synthase responsible for the synthesis of amylose, in 400 KNO_3_^+^ plants. It was previously established for rice seeds that salt stress leads to a decrease in amylose biosynthesis [66,67]. However, another study [68] found a relative increase of amylose for seeds of triticale wheat under salt stress. In any case, despite the differences between the two plant groups, overall, these results suggest that the robustness of the stress response system in *C. glauca* is regulated by a limited number of genes that tightly regulate detoxification and protein/enzyme stability.

One of the mechanisms used by *C. glauca* to cope with salinity is the reduction of growth ensuring relative high values of hydration (as evaluated by the branchlets relative water content, RWC), ion homeostasis, and photosynthetic functioning [22]. In this context, the down-regulation of genes involved in growth and developmental processes, e.g., histones (DNA), patellins (cell polarity and patterning) [69], or cysteine proteases such as XCP1 (differentiation of xylem vessels) [70], is not surprising. In addition, previous studies on the effects of salinity stress have often shown negative regulation of genes involved in cell wall biosynthesis, a part of the growth process, or up-regulation of genes involved in cell wall degradation [40,71]. Accordingly, in this study, we found down-regulation of a set of cell-wall related genes, e.g., pectin esterase/pectin esterase inhibitor (400 NOD^+^), a polygalacturonase (400 KNO_3_^+^), an arabinogalactan (400 NOD^+^ and 400 KNO_3_^+^), a cellulose synthase (600 KNO_3_^+^), a laccase, (600 NOD^+^ and 600 KNO_3_^+^), and a regulator of cell expansion, cobra-like protein (600 NOD^+^).

As mentioned above, the expression of several genes encoding TFs putatively involved in signal transduction was differentially regulated under salt stress. The striking effect here, however, was the consistent downregulation of RADIALIS-LIKE SANT/MYB TFs (RADIALIS-LIKE 3 and RADIALIS-LIKE5) in both NOD^+^ and KNO_3_^+^ plants (400 and 600 mM NaCl). These TFs form a small family in *Arabidopsis* [72], and one of them (RSM1), which is down-regulated under salinity, was recently shown to modulate seedling development in response to ABA and salinity [73]. The function of RSM1 in the response to long-term salinity stress was confirmed in Castor Bean (*Ricinus communis*) by Han et al. [35], who suggested that the downregulation of RSM1 by salt was regulated via epigenetic modifications of the promoter.

Hierarchical cluster analysis identified 10 (KNO_3_^+^) and 11 (NOD^+^) gene clusters (Figure 4 and Figure 5), all including genes associated with binding, catalytic activity, transport, signaling, defense against biotic and abiotic stress, and response to stimuli. Among these, five clusters (C0–C2, C8–C9) were sharply downregulated in KNO_3_^+^ plants exposed to 600 mM NaCl, while in NOD^+^ plants the number of negatively regulated gene clusters was seven (C0-C5, and C10). Despite the differences between the two plant groups, likely related to the nitrogen source [23,24], the overall set of results shows that *C. glauca* is able to balance the transcriptional activity at 200 mM NaCl to levels similar to that of the control (Figure 4 and Figure 5). At 400 mM NaCl the expression patterns tend to be more variable, while at 600 mM NaCl transcript levels are severely affected, i.e., ca. 60 % of the DEGs are downregulated in both KNO_3_^+^ and NOD^+^ plants. Indeed, PPI networks of DEGs were found only at 600 mM NaCl in both plant groups, mainly related to the response to stress and stimuli, as well as to cell death. In our view, despite the positive regulation of some stress-related genes, at the highest salt concentration the plant is unable to sustain the necessary transcriptional activity to cope with salinity. Comparing this trend with the overall impact of 600 mM NaCl on plant growth, eco-physiological performance, proteome, and metabolome [22,24,25,26,27,30], it seems clear that under the experimental conditions used, this *C. glauca* ecotype tolerates at least 400 mM. Such tolerance is related to a constitutive defense system, an enhanced antioxidative and osmoprotectant status, as well as a tight mechanism of transcription regulation and post-transcriptional modification.

Finally, codon and amino acid usage of DEGs among KNO_3_^+^ and NOD^+^ plants revealed a distinct pattern. Compositional constrain and translational efficiency were found to be the major driving force in maintaining the codon usage indices among DEGs of NOD^+^ plants. In this group, the presence of costly proteins with higher amounts of aromatic amino acids highlights the importance of energy rich amino acids. Moreover, COG analysis made it evident that in NOD^+^ plants, the signaling proteins were mostly functional. This may be due to a strong influence exerted by the symbiosis with *Frankia casuarinae* Thr. While the bacteria inside nodules are mostly dead at 200 mM NaCl [23], the effect of nodulation on the plant should still be in place. In legumes, extensive research has revealed complex local and systemic control of root architecture during nodulation [74,75]. Consistent with these data, legume studies have shown that nodulation affects the response to salt stress with regard to pathways involved in photosynthesis, respiration, anion transport, and plant defense [40]. Similar processes have evolved in actinorhizal plants [76]. However, they were not yet analyzed in *Casuarina* spp. In short, it is not surprising that even if the microsymbionts in nodules do not survive salt stress, the symbiosis still affects the stress response.

In conclusion, the results highlight the complexity of the molecular interactions leading to salinity tolerance in *C. glauca* likely linked to long-term ecological adaptation and, in this case, independent of symbiotic *Frankia*.

## 4. Materials and Methods

### 4.1. Growing Conditions and Experimental Setup

*Casuarina glauca* clones were grown in Broughton and Dillworth’s (BD) medium under the conditions previously described by Zhong et al. [77] and Tromas et al. [78]. Six-month-old plants were either nodulated by *Frankia casuarinae* Thr (NOD^+^) or supplemented with mineral nitrogen (KNO_3_^+^) and maintained in a walk-in growth chamber (10,000 EHHF, ARALAB, Portugal) under environmental controlled conditions of temperature (26/22 °C), photoperiod (12 h), relative humidity (70%), external CO_2_ levels (380 μL L^−1^), and irradiance (ca. 500 μmol m^−2^s^−1^). Salt stress was gradually imposed through the addition of 50 mM NaCl per week to the nutrient solution until concentrations of 200, 400 and 600 mM were obtained, respectively. Control plants were supplemented only with mineral nutrients without the addition of NaCl. In all cases, the nutrient solution was renewed twice per week. To ensure the same age of all plants at the time of sample collection, i.e., one week after the exposure to each NaCl concentration, stress implementation was imposed sequentially from the highest to the lowest salt condition, i.e., the 600 mM NaCl group was treated first, followed by the 400 mM group (when the 600 mM group surpassed 200 mM NaCl), and the 200 mM group (when the 400 mM group surpassed 200 mM NaCl). Branchlets from four plants per treatment were harvested, frozen immediately in liquid nitrogen, and stored at –80 ºC until RNA extraction.

### 4.2. Total RNA Extraction and Library Preparation

Total RNA from branchlets of NOD^+^ and KNO_3_^+^ plants exposed to the three stress levels, as well as from control plants was extracted using the GeneJet Plant RNA Purification Kit (Thermo Scientific, Waltham, MA, USA) and digested with dsDNAse (Thermo Scientific, USA) to remove genomic DNA, as per manufacturer’s instructions. The quality of the RNA samples was verified by electrophoresis in 1% agarose–Tris acetate EDTA buffer containing GelRed Nucleic Acid Gel Stain (Biotium, Fremont, CA, USA), by evaluating the integrity of the 28S and 18S ribosomal RNA bands and absence of smears, as well as with a bioanalyzer (Agilent 2100, Agilent, Santa Clara, CA, USA), by determining the RNA integrity number (RIN > 8.5). Libraries were prepared with the TruSeq RNA Sample Prep Kit v2 (Illumina, San Diego, CA, USA) and sequenced on an Illumina HiSeq 2000 platform (2 × 125 bp pair-end reads; 30 million reads per sample) at Macrogen (Seul, South Korea). Raw data is available in NCBI Sequence Read Archive (SRA) BioProject SymbSaltStress under accession PRJNA706159 (https://www.ncbi.nlm.nih.gov/sra/PRJNA706159, accessed on 1 September 2022).

### 4.3. Processing and Mapping of Illumina Reads

The sequenced raw reads were assessed for integrity, quality and contamination through the application of the following tools: FastQC version 0.11.8 to analyze reads quality [79] and FastQ Screen version 0.13 to survey putative contaminants, run against the genome of 14 default pre-indexed species and adaptors [80]. Trimmomatic version 0.39.1 was then used to eliminate the remaining adaptors and low-quality or small reads [81], using ILLUMINACLIP with keepBothReads option, SLIDINGWINDOW:4:15, MAXINFO:36:0.5 and MINLEN:36. After filtering and trimming, Trinity version 0.39.1 was used to perform de novo transcriptome assembly, combining all samples to generate one single assembly [82]. This software was developed specifically for short reads and is advantageous for non-model plant sequence assemblies [83]. The assembled transcriptome was assessed for completeness through the gVolante [84] online interface with BUSCO v2.0.1 option [85]. To align the reads against the assembled transcriptome, the sequences were processed with Trinity tool Bowtie2 version 2.3.5 [86,87] and the aligned reads for each condition were quantified at gene expression level with RSEM version 1.3.2. [88]. The normalized expression of all samples was estimated using the Trimmed Mean of M values (TMM). A principal component analysis (PCA) was performed to survey the relatedness of all samples using the function plotPCA in R studio version 3.6.0 [89].

### 4.4. Identification of Differentially Expressed Genes

To identify differentially expressed genes (DEGs), a set of R packages were applied, namely edgeR version 3.26.8 [90], DESeq version 1.36.0 [91] and NOISeq version 2.28.0 [92]. DESeq fits a generalized linear model to estimate variance-mean dependence in count data, testing for differential expression based on the negative binomial distribution. The dispersion was estimated through blind mode [91]. On the other hand, edgeR uses weighted likelihood methods to implement a flexible empirical Bayes approach to allow gene-specific variation estimates even with very few or no replicates [93]. The dispersion value, which must be fixed manually in the absence of replicates, was set to 0.1. By contrast, NOISeq is an exploratory analysis that tests for differential expression between two experimental conditions without parametric assumptions that can simulate technical replicates. This method relies on the premise that read counts follow a multinomial distribution, where probabilities for each feature are the probability of a read to map to it [92]. Differential expression was computed using a stringent threshold of q = 0.9 along with the following parameters: pnr = 0.2, nss = 5, v = 0.02.

To study the effect of salinity, these three tools were used to identify DEGs at each salinity level (200, 400, and 600 mM NaCl) compared to the control (0 mM NaCl), for KNO_3_^+^ and NOD^+^ plants, respectively. The results were adjusted with the Benjamini and Hochberg’s approach for controlling the false discovery rate (FDR) [94]. A filter of FDR < 0.05 and normalized log2 fold change (log2FC) > |2| were set to define DEGs. DEGs commonly detected by edgeR, DESeq and NOISeq were combined to increase the accuracy of results, and only the genes detected as differentially expressed by all the three tools were used in downstream analyses. To visualize the resulting expression profiles, volcano plots, heatmaps, barplots, and Venn diagrams were prepared for each combination, using stats and graphics core R packages and Python’s Matplotlib 3.2.1 library [95].

The Clust version 1.8.10 command-line tool was applied to visualize the expression patterns of the detected DEGs and to find co-expressed genes [96]. This is a fully automated method for the identification of gene clusters that are co-expressed in heterogeneous datasets that automatically normalizes the data and determines the best number of clusters, based on a selected tightness, which in this case was five (5). Then, the results were filtered to keep only the DEGs that were found uniquely in one of the clusters and to find terms related to salt stress response.

### 4.5. Functional Annotation and Enrichment Analysis

The Basic Local Alignment Search Tool (BLAST) version 2.9.0 command line application from the NCBI C++ Toolkit [97] was used for functional annotation of DEGs, using homology searches with the latest references, comprising only the expertly curated component of UniProtKB. Results were filtered by maximum E-Value of 1.0E-3 and minimum Identities of 40% [98]. The resultant annotated proteins were then characterized by Gene Ontology (GO) terms: Cellular Component (CC), Molecular Function (MF), and Biological Process (BP), using the Uniprot and QuickGO APIs [99] to retrieve direct terms and GO term ancestors.

An over-representation analysis (ORA) was implemented by g:GOSt functional profiling tool from gProfiler website, which was applied using g:SCS tailored algorithm that uses a minimum hypergeometric test (Fisher’s exact test). ShinyGO version 0.61 webtool, which is also based on hypergeometric distribution followed by FDR correction was used to retrieve Protein-Protein Interactions (PPIs) from the STRING database [100]. In both tools, Arabidopsis thaliana was selected as the organism of interest and separate log2FC ranked lists of DEGs for each salinity condition were used as inputs.

### 4.6. Downstream Bioinformatics Analysis

The downstream analysis included the codon and amino acid usage analysis of DEGs. Compositional constrain (GC and GC3), effective number of codons (ENc), and frequency of optimal codons (Fop) were determined using CodonW software. Biosynthetic energy costs of DEGs were obtained through DAMBE software. The Cluster of Orthologue analysis (COG) was performed using an in-house python-based program where information regarding different COG families was taken from the COG database (http://www.ncbi.nlm.nih.gov/COG, accessed on 1 September 2022). A detailed study on biological networks maintained by DEGs was performed.

## Figures and Tables

**Figure 1 plants-11-02942-f001:**
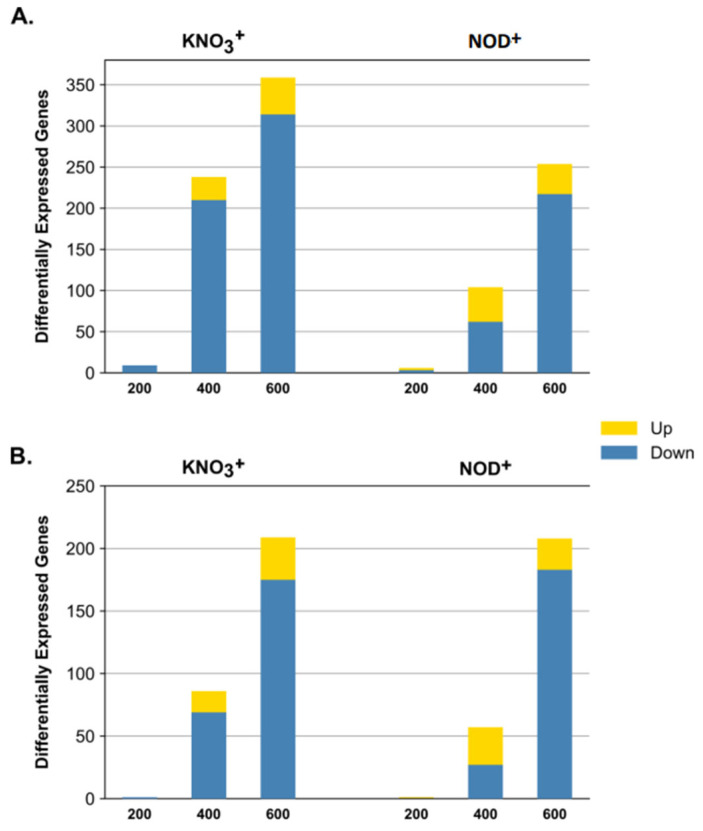
Differential gene expression in branchlets of *Casuarina glauca* plants nodulated by nitrogen-fixing *Frankia casuarinae* Thr (NOD^+^) or supplied with KNO_3_ (KNO_3_^+^). 200: 200 vs. 0 mM NaCl; 400: 400 vs. 0 mM NaCl; and 600: 600 vs. 0 mM NaCl, with a False Discovery Rate < 0.05. (**A**) Total number of DEGs. (**B**) Number of DEGs specific to each salinity condition relative to control.

**Figure 2 plants-11-02942-f002:**
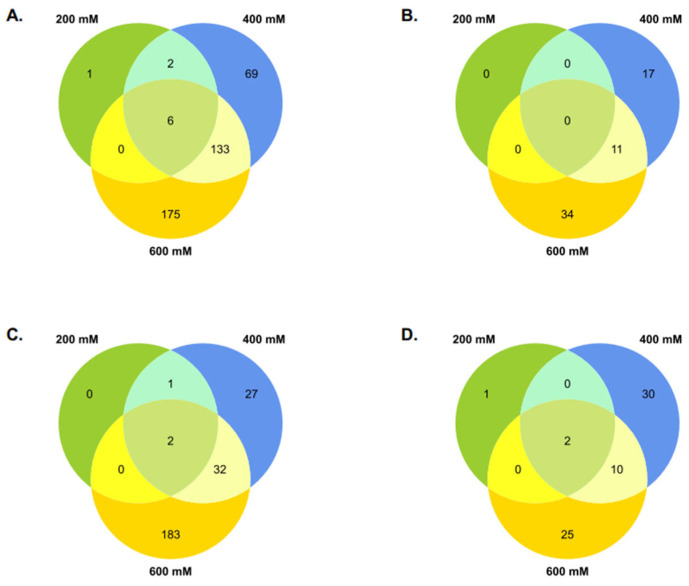
Differential gene expression in *Casuarina glauca* plants supplied with KNO_3_ (KNO_3_^+^) or nodulated by nitrogen-fixing *Frankia casuarinae* Thr (NOD^+^), grown at 200 mM, 400 mM, and 600 mM NaCl. Total number of treatment-specific differentially expressed genes (DEGs) for each salt treatment relative to the control (0 mM NaCl). 200: 200 vs. 0 mM NaCl; 400: 400 vs. 0 mM NaCl; and 600: 600 vs. 0 mM NaCl. (**A**) KNO_3_^+^ downregulated DEGs. (**B**) KNO_3_^+^ upregulated DEGs. (**C**) NOD^+^ downregulated DEGs. (**D**) NOD^+^ upregulated DEGs.

**Figure 3 plants-11-02942-f003:**
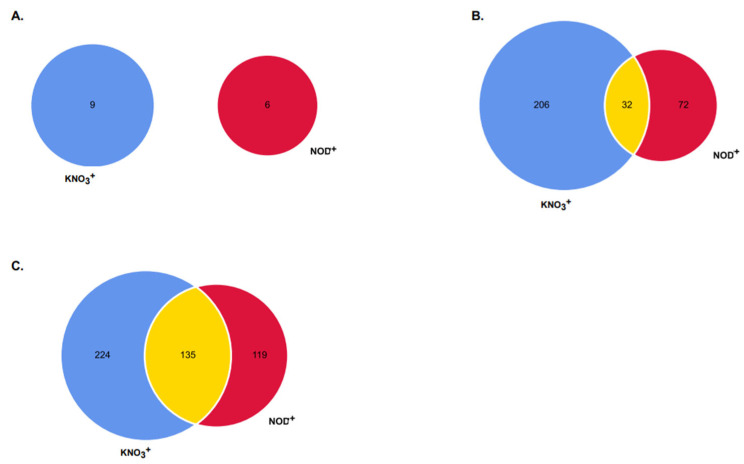
Specific and overlapping genes between *Casuarina glauca* plants supplied with KNO_3_ (KNO_3_^+^) or nodulated by nitrogen-fixing *Frankia casuarinae* Thr (NOD^+^). Significant differentially expressed genes (DEGs), with a False Discovery Rate (FDR) < 0.05, were detected between the control (0 mM NaCl) and salinity-exposed plants: (**A**) 200 mM NaCl; (**B**) 400 mM NaCl; (**C**) 600 mM NaCl.

**Figure 4 plants-11-02942-f004:**
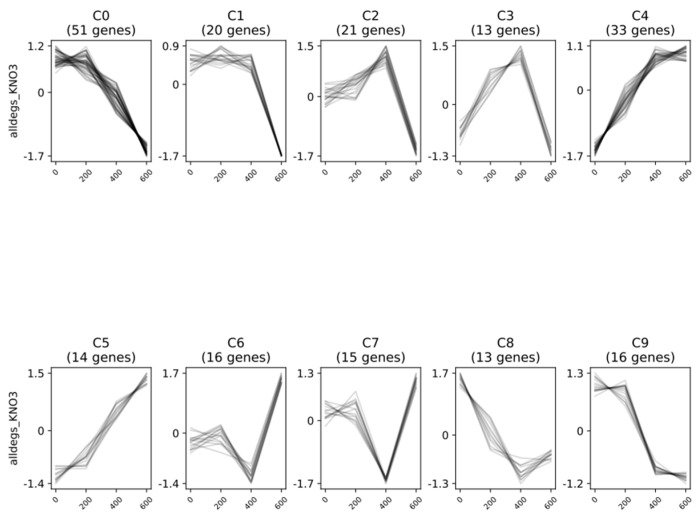
Pattern of expression of differentially expressed genes (DEGs) in non-nodulated *Casuarina glauca* plants supplied with KNO_3_ (KNO_3_^+^), grouped by predicted co-expressed gene clusters. Clustering analysis was performed by Clust tool with normalized reads from samples grown at 0, 200, 400, and 600 mM NaCl.

**Figure 5 plants-11-02942-f005:**
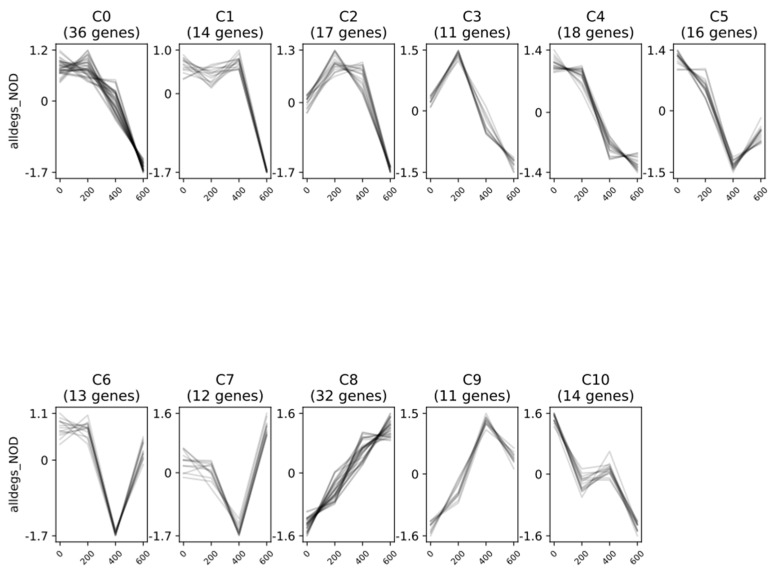
Pattern of expression of differentially expressed genes (DEGs) in *Casuarina glauca* nodulated by nitrogen-fixing *Frankia casuarinae* Thr (NOD^+^), grouped by predicted co-expressed gene clusters. Clustering analysis was performed by Clust tool, with normalized reads from samples grown at 0, 200, 400, and 600 mM NaCl.

**Figure 6 plants-11-02942-f006:**
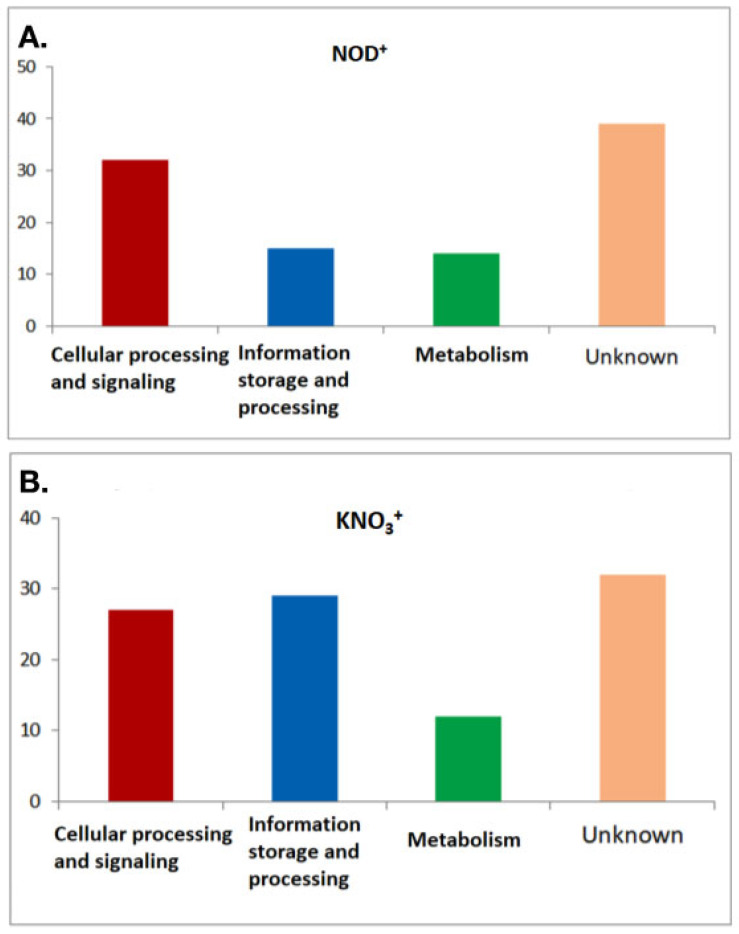
COG distribution pattern in NOD^+^ plants (**A**) and non-nodulated KNO_3_^+^ plants (**B**).

**Table 1 plants-11-02942-t001:** Differential gene expression quantification in branchlets of *Casuarina glauca* nodulated by nitrogen-fixing *Frankia casuarinae* Thr (NOD^+^) or non-nodulated plants supplied with KNO_3_ (KNO_3_^+^), grown at control (0 mM NaCl) and three salinity conditions (200 mM, 400 mM and 600 mM NaCl). Number of total expressed genes by each plant, at each condition, number of genes expressed by both treatment and control of each comparison (common genes percentage relative to the respective average gene expression between brackets), number of differential expressed genes (DEGs) detected by DESeq, NOISeq and edgeR analyses, number of overlapping DEGs in the three analyses (DEGs percentages relative to the respective average gene expression between brackets) and number of respective annotated DEGs, enumerated by type of regulation. Detected and overlapping DEGs correspond to the number of significant DEGs in each salinity-stress treatment in comparison with the control NaCl (respectively, 200: 200 mM vs. 0 mM, 400: 400 mM vs. 0 mM and 600: 600 mM vs. 0 mM), with a False Discovery Rate < 0.05 and a normalized log2 fold change (log2FC) > 2 or <−2.

Plant Series	[NaCl] mM	Expressed Genes			Detected DEGs			Overlapping DEGs					
		Stressed	Control	Common	DESeq	NOISeq	edgeR	All			Annotated		
								Total (%)	Up	Down	Total (%)	Up	Down
**KNO_3_^+^**	200	20,765	19,913	15,928 (78%)	58	330	23	9 (0.04%)	0	9	2 (22%)	0	2
	400	19,078	19,913	15,079 (77%)	524	1213	381	238 (1%)	28	210	125 (53%)	8	117
	600	19,092	19,913	14,976 (77%)	650	1450	612	359 (2%)	45	314	164 (46%)	18	146
**NOD^+^**	200	19,397	20,278	16,386 (83%)	44	204	20	6 (0.03%)	3	3	4 (67%)	3	1
	400	20,470	20,278	16,137 (79%)	359	962	184	104 (0.5%)	42	62	49 (47%)	21	28
	600	19,472	20,278	15,547 (78%)	548	1261	373	254 (1%)	37	217	111 (44%)	14	97

**Table 2 plants-11-02942-t002:** Top 10 most down and upregulated differentially expressed genes (DEGs) in non-nodulated *Casuarina glauca* plants supplied with KNO_3_ (KNO_3_^+^), grown at 400 mM NaCl relative to the control (0 mM NaCl), ordered by log2 fold-change (log2FC). NBCI gene symbol, protein name, log2FC and respective Gene Ontology (GO) molecular function(s) and biological process(es) were retrieved from UniprotKB database. Protein names coloured in blue are also among the top 10 most down or upregulated DEGs, respectively, at 400 mM NaCl in plants nodulated by nitrogen-fixing *Frankia casuarinae* Thr (NOD^+^).

Gene	Protein Name	Log_2_FC	Molecular Function/Biological Process
**Downregulated**			
*XCP1*	Cysteine protease XCP1	−6.883	Programmed cell death; proteolysis
*AT3G10080*	Germin-like protein subfamily 3 member 2	−6.658	Manganese ion binding; nutrient reservoir activity
*ACA13*	Putative calcium-transporting ATPase 13, plasma membrane-type	−6.476	ATP, calmodulin and metal ion binding; ATPase-coupled cation and calcium transmembrane transporter activity
*RL5*	Protein RADIALIS-like 5	−5.931	DNA binding
*ERF020*	Ethylene-responsive transcription factor ERF020	−5.644	Ethylene-activated signaling pathway; response to chitin
*YAH3*	Histone H3.2	−5.476	DNA binding; protein heterodimerization activity
*LRX2*	Leucine-rich repeat extensin-like protein 2	−5.157	Cell wall organization
*PATL4*	Patellin-4	−5.129	Cell cycle and division; cellular response to auxin stimulus; auxin polar transport
*PGL3*	Polygalacturonase 1 beta-like protein 3	−5.073	Cell size determination
*FLA12*	Fasciclin-like arabinogalactan protein 12	−5.055	Plant-type secondary cell wall biogenesis
**Upregulated**			
*WAXY*	Granule-bound starch synthase 1, chloroplastic/amyloplastic	3.459	Starch biosynthetic process
*HSP70*	Heat shock 70 kDa protein	3.363	Cellular response to unfolded protein; chaperone cofactor-dependent protein refolding
*N/A*	Myrcene synthase, chloroplastic	3.129	Magnesium ion binding; myrcene synthase activity
*SDI1*	Protein sulfur deficiency-induced 1	3.000	Cellular response to sulfur starvation; regulation of glucosinolate biosynthetic process and sulfur utilization
*UGT87A1*	UDP-glycosyltransferase 87A1	2.824	UDP-glycosyltransferase activity
*ARF16*	Auxin response factor 16	2.644	Auxin-activated signaling pathway; cell division; pattern specification process; response to auxin; root cap development
*LECRK111*	Putative L-type lectin-domain containing receptor kinase I.11	2.369	Defense response to bacterium and oomycetes
*TOGT1*	Scopoletin glucosyltransferase	2.290	Identical protein binding; scopoletin glucosyltransferase activity; UDP-glycosyltransferase activity

**Table 3 plants-11-02942-t003:** Top 10 most down and upregulated differentially expressed genes (DEGs) in non-nodulated *Casuarina glauca* plants supplied with KNO_3_ (KNO_3_^+^), grown at 600 mM NaCl relative to the control (0 mM NaCl), ordered by log2 fold-change (log2FC). NBCI gene symbol, protein name, log2FC and respective Gene Ontology (GO) molecular function(s) and biological process(es) retrieved from UniprotKB database. Protein names coloured in blue are also on the top 10 most down or upregulated DEGs, respectively, at 600 mM NaCl in plants nodulated by nitrogen-fixing *Frankia casuarinae* Thr (NOD^+^).

Gene	Protein Name	Log_2_FC	Molecular Function/Biological Process
**Downregulated**			
*XCP1*	Cysteine protease XCP1	−6.883	Programmed cell death; proteolysis
*AT3G61750*	Cytochrome b561 and DOMON domain-containing protein At3g61750	−6.775	Metal ion binding
*N/A*	Lectin	−6.600	Mannose binding
*LAC22*	Laccase-22	−6.459	Iron ion homeostasis and transport; lignin catabolic process
*N/A*	Seed lectin	−6.392	Carbohydrate binding; metal ion binding
*CESA9*	Cellulose synthase A catalytic subunit 9 [UDP-forming]	−6.109	Cell wall organization and biogenesis
*SERK4*	Somatic embryogenesis receptor kinase 4	−5.954	Cell death; phosphorylation; response to hypoxia and chitin; leaf senescence and seedling development
*RL5*	Protein RADIALIS-like 5	−5.931	DNA binding
*LECRK91*	L-type lectin-domain containing receptor kinase IX.1	−5.781	Defense response to bacterium and oomycetes; positive regulation of cell death and hydrogen peroxide metabolic process
*LRK10L-1.4*	Leaf rust 10 disease-resistance locus receptor-like protein kinase-like 1.4	−5.700	Cellular response to hypoxia; phosphorylation
**Upregulated**			
*PLT6*	Probable polyol transporter 6	4.555	Glucose import
*MAKR6*	Probable membrane-associated kinase regulator 6	4.366	Abscisic acid-activated signaling pathway
*UGT709C2*	7-deoxyloganetic acid glucosyltransferase	3.415	(-)-secologanin biosynthetic process
*SAT*	Stemmadenine O-acetyltransferase	3.392	Alkaloid metabolic process
*DTX56*	Protein DETOXIFICATION 56	3.392	Cellular response to carbon dioxide; regulation of stomatal opening
*HSP70*	Heat shock 70 kDa protein	3.123	Cellular response to unfolded protein; chaperone cofactor-dependent protein refolding
*N/A*	Chitinase 2	2.807	Chitin catabolic process
*TOGT1*	Scopoletin glucosyltransferase	2.713	Identical protein binding; UDP-glycosyltransferase activity
*DTX27*	Protein DETOXIFICATION 27	2.690	Antiporter activity; xenobiotic transmembrane transporter activity
*CNGC4*	Cyclic nucleotide-gated ion channel 4	2.672	Plant-type hypersensitive response

**Table 4 plants-11-02942-t004:** Top 10 most down and upregulated differentially expressed genes (DEGs) in *Casuarina glauca* nodulated by nitrogen-fixing *Frankia casuarinae* Thr (NOD^+^), grown at 400 mM NaCl relative to the control (0 mM NaCl), ordered by log2 fold-change (log2FC). NBCI gene symbol, protein name, log2FC and respective Gene Ontology (GO) molecular function(s) and biological process(es) retrieved from UniprotKB database. Protein names coloured in blue are also among the top 10 most down or upregulated DEGs, respectively, at 400 mM NaCl in non-nodulated plants supplied with KNO_3_ (KNO_3_^+^).

Gene	Protein Name	Log_2_FC	Molecular Function/Biological Process
**Downregulated**			
*RL5*	Protein RADIALIS-like 5	−7.794	DNA binding
*CCD8*	Carotenoid cleavage dioxygenase 8, chloroplastic	−7.249	Auxin polar transport; carotene catabolic process; leaf morphogenesis; response to auxin; secondary shoot formation; strigolactone biosynthetic process; xanthophyll catabolic process
*PER64*	Peroxidase 64	−5.977	Hydrogen peroxide catabolic process; response to oxidative stress
*PME6*	Probable pectinesterase/pectinesterase inhibitor 6	−5.274	Cell wall modification; pectin catabolic process
*GLR2.8*	Glutamate receptor 2.8	−5.247	Calcium ion transport; calcium-mediated signaling; cellular response to amino acid stimulus
*FLA12*	Fasciclin-like arabinogalactan protein 12	−4.542	Plant-type secondary cell wall biogenesis
*RL3*	Protein RADIALIS-like 3	−4.261	DNA binding; DNA-binding transcription factor activity
*COBL4*	COBRA-like protein 4	−4.087	Cellulose microfibril organization; plant-type cell wall biogenesis; plant-type cell wall cellulose biosynthetic process; plant-type cell wall organization; plant-type secondary cell wall biogenesis
*YAH3*	Histone H3.2	−3.906	DNA binding; protein heterodimerization activity
*ROMT*	Trans-resveratrol di-O-methyltransferase	−3.729	Aromatic compound biosynthetic process; methylation
**Upregulated**			
*GDH2*	Glutamate dehydrogenase 2	5.087	Glutamate catabolic process; response to cadmium ion
*CYP707A2*	Abscisic acid 8’-hydroxylase 2	4.321	Abscisic acid catabolic process; abscisic acid metabolic process; oxidation-reduction process; release of seed from dormancy; response to red light; response to red or far red light; sterol metabolic process
*MYBAS1*	Myb-related protein MYBAS1	4.129	Regulation of transcription, DNA-templated
	Pathogenesis-related thaumatin-like protein 3.5	3.954	Defense response; response to abscisic acid; response to biotic stimulus
*DTX27*	Protein DETOXIFICATION 27	3.762	Antiporter activity; transmembrane transporter activity; xenobiotic transmembrane transporter activity
*EPS1*	Protein enhanced pseudomonas susceptibility 1	3.654	Defense response; regulation of defense response to bacterium; regulation of defense response to fungus; response to bacterium; response to jasmonic acid; salicylic acid biosynthetic process
*PEPR1*	Leucine-rich repeat receptor-like protein kinase PEPR1	3.584	Immune response; innate immune response; response to jasmonic acid; Response to wounding
*UGT73E1*	UDP-glycosyltransferase 73E1	3.523	Transferase activity, transferring hexosyl groups
*SDI1*	Protein sulfur deficiency-induced 1	3.459	Cellular response to sulfur starvation; regulation of glucosinolate biosynthetic process; regulation of sulfur utilization
*HSP70*	Heat shock 70 kDa protein	3.301	Cellular response to unfolded protein; chaperone cofactor-dependent protein refolding; protein refolding; response to unfolded protein

**Table 5 plants-11-02942-t005:** Top 10 most down and upregulated differentially expressed genes (DEGs) in *Casuarina glauca* nodulated by nitrogen-fixing *Frankia casuarinae* Thr (NOD^+^), grown at 600 mM NaCl relative to the control (0 mM NaCl), ordered by log2 fold-change (log2FC). NBCI gene symbol, protein name, log2FC and respective Gene Ontology (GO) molecular function(s) and biological process(es) retrieved from UniprotKB database. Protein names coloured in blue are also among the top 10 most down or upregulated DEGs, respectively, at 600 mM NaCl in non-nodulated plants supplied with KNO_3_ (KNO_3_^+^).

Gene	Protein Name	Log_2_FC	Biological Process
**Downregulated**			
*RL5*	Protein RADIALIS-like 5	−7.794	DNA binding
*CCD8*	Carotenoid cleavage dioxygenase 8, chloroplastic	−6.134	auxin polar transport; carotene catabolic process; leaf morphogenesis; response to auxin; secondary shoot formation; strigolactone biosynthetic process Xanthophyll catabolic process
*TAO1*	Disease resistance protein TAO1	−5.727	defense response to bacterium; defense response to bacterium, incompatible Interaction; signal transduction
*N/A*	Lectin	−5.643	Mannose binding
*pod*	Peroxidase 15	−5.529	Hydrogen peroxide catabolic process; response to oxidative stress
*GLR2.8*	Glutamate receptor 2.8	−5.247	calcium ion transport; calcium-mediated signaling; cellular response to amino Acid stimulus
*LAC22*	Laccase-22	−5.183	Iron ion homeostasis; iron ion transport; lignin catabolic process
*COBL4*	COBRA-like protein 4	−5.087	cellulose microfibril organization; plant-type cell wall biogenesis; plant-type cell wall cellulose biosynthetic process; plant-type cell wall organization; plant-type Secondary cell wall biogenesis
*LECRK91*	L-type lectin-domain containing receptor kinase IX.1	−5.087	defense response; defense response to bacterium; defense response to oomycetes; positive regulation of cell death; positive regulation of hydrogen peroxide Metabolic process
*LRK10L-1.4*	Leaf rust 10 disease-resistance locus receptor-like protein kinase-like 1.4	−5.058	Cellular response to hypoxia; phosphorylation
**Upregulated**			
*At3g18200*	WAT1-related protein At3g18200	2.906	Transmembrane transporter activity
*TIC214*	Protein TIC 214	2.906	Protein transport
*SAT*	Stemmadenine O-acetyltransferase	2.700	Alkaloid metabolic process
*DTX56*	Protein DETOXIFICATION 56	2.929	Cellular response to carbon dioxide; regulation of stomatal opening
*MAKR6*	Probable membrane-associated kinase regulator 6	2.861	Abscisic acid-activated signaling pathway
*HSP70*	Heat shock 70 kDa protein	2.728	Cellular response to unfolded protein; chaperone cofactor-dependent protein refolding; protein refolding; response to unfolded protein
*HSP22.7*	22.7 kDa class IV heat shock protein	2.357	Endoplasmic reticulum lumen
*SDI1*	Protein sulfur deficiency-induced 1	4.087	Cellular response to sulfur starvation; regulation of glucosinolate biosynthetic process; regulation of sulfur utilization
*PEPR1*	Leucine-rich repeat receptor-like protein kinase PEPR1	4.584	Immune response; innate immune response; response to jasmonic acid; response to wounding
*At1g48100*	Polygalacturonase At1g48100	2.495	Carbohydrate metabolic process; plant-type cell wall modification involved in multidimensional cell growth

**Table 6 plants-11-02942-t006:** Significantly enriched Gene Ontology (GO) terms, identified by Gene Set Enrichment Analysis (GSEA) of downregulated differentially expressed genes (DEGs) in non-nodulated *Casuarina glauca* plants supplied with KNO_3_ (KNO_3_^+^) or nodulated by nitrogen-fixing *Frankia casuarinae* Thr (NOD^+^), grown at 400 mM NaCl, relative to control (0 mM NaCl). GO aspect, GO ID, GO Name, False Discovery Rate (FDR), intersection size, effective domain size and main gene intersections were retrieved from g:GOSt functional profiling of the gProfiler website, within a threshold of FDR < 0.05.

Aspect	GO ID	GO Name	FDR	Intersection Size	Effective Domain Size	Intersections
**KNO_3_^+^**						
**MF**	GO:0008194	UDP-glycosyltransferase activity	0.02845	6	21,106	GATL2
**BP**	GO:0071555	cell wall organization	0.00081	9	23,362	LRX2
	GO:0045229	external encapsulating structure organization	0.00134	9	23,362	LRX2
	GO:0050829	defense response to Gram-negative bacteria	0.00667	3	23,362	EDS1
	GO:0060866	leaf abscission	0.02170	2	23,362	EDS1
	GO:0000271	polysaccharide biosynthetic process	0.04131	5	23,362	GATL2
**CC**	GO:0071944	cell periphery	0.00013	24	21,026	XCP1
	GO:0046658	anchored component of plasma membrane	0.01625	4	21,026	FLA10
**NOD^+^**						
**MF**	GO:0004352	glutamate dehydrogenase (NAD^+^) activity	0.01952	1	21,106	GDH2
**BP**	GO:0016115	terpenoid catabolic process	0.02074	2	23,362	CYP707A2
**CC**	GO:0005886	plasma membrane	0.00174	10	21,026	GDH2
	GO:0071944	cell periphery	0.00842	10	21,026	GDH2

**Table 7 plants-11-02942-t007:** Significantly enriched Gene Ontology (GO) terms, identified by Gene Set Enrichment Analysis (GSEA) of downregulated differentially expressed genes (DEGs) in non-nodulated *Casuarina glauca* plants supplied with KNO_3_ (KNO_3_^+^) or nodulated by nitrogen-fixing *Frankia casuarinae* Thr (NOD^+^), grown at 600 mM NaCl, relative to control (0 mM NaCl). GO aspect, GO ID, GO Name, False Discovery Rate (FDR), intersection size, effective domain size and main gene intersections were retrieved from g:GOSt functional profiling of the gProfiler website, within a threshold of FDR < 0.05. GO Names coloured in blue were found to be enriched for DEGs at 600 mM NaCl in both plant series.

Aspect	GO ID	GO Name	FDR	Intersection Size	Effective Domain Size	Intersections
**KNO_3_^+^**						
**MF**	GO:0004674	protein serine/threonine kinase activity	0.00289	11	21,106	STN8
	GO:0005524	ATP binding	0.01405	17	21,106	AT2G40270
	GO:0035639	purine ribonucleoside triphosphate binding	0.01480	18	21,106	AT2G40270
	GO:0032559	adenyl ribonucleotide binding	0.02552	17	21,106	AT2G40270
	GO:0008144	drug binding	0.03319	17	21,106	AT2G40270
	GO:0097367	carbohydrate derivative binding	0.03632	18	21,106	AT2G40270
**BP**	GO:0006468	protein phosphorylation	0.00152	13	23,362	AT2G40270
	GO:0009814	defense response, incompatible interaction	0.02407	5	23,362	RPP1
	GO:0009838	abscission	0.04514	3	23,362	SAG101
**NOD^+^**						
**MF**	GO:0004674	protein serine/threonine kinase activity	0.00439	9	21,106	CCR4
	GO:0005524	ATP binding	0.00930	14	21,106	TAO1
	GO:0102251	all-trans-beta-apo-10’-carotenal cleavage oxygenase activity	0.01331	1	21,106	CCD8
	GO:0102396	9-cis-10’-apo-beta-carotenal cleavage oxygenase activity	0.01331	1	21,106	CCD8
	GO:0032559	adenyl ribonucleotide binding	0.01575	14	21,106	TAO1
	GO:0008144	drug binding	0.01988	14	21,106	TAO1
	GO:0004620	phospholipase activity	0.02127	3	21,106	AT1G06800
	GO:0035639	purine ribonucleoside triphosphate binding	0.03000	8	21,106	TAO1
	GO:0032555	purine ribonucleotide binding	0.04092	8	21,106	TAO1
**BP**	GO:0051704	multi-organism process	0.00033	13	23,362	TAO1
	GO:0042742	defense response to bacterium	0.00076	7	23,362	TAO1
	GO:0006468	protein phosphorylation	0.00869	10	23,362	CCR4
	GO:0031349	positive regulation of defense response	0.00903	4	23,362	SOBIR1
	GO:0016042	lipid catabolic process	0.01363	5	23,362	CCD8
	GO:0060866	leaf abscission	0.01451	2	23,362	SAG101
	GO:0008219	cell death	0.01457	5	23,362	SOBIR1
	GO:0009814	defense response, incompatible interaction	0.03085	4	23,362	TAO1
	GO:0001666	response to hypoxia	0.04357	5	23,362	PLP2

## Data Availability

Raw data is available in NCBI Sequence Read Archive (SRA) BioProject SymbSaltStress under accession PRJNA706159 (https://www.ncbi.nlm.nih.gov/sra/PRJNA706159, accessed on 1 September 2022).

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
