# Peer review of "Salt Stress Tolerance in Casuarina glauca: Insights from the Branchlets Transcriptome"

_plants, 2022, doi:10.3390/plants11212942_

Round 1
Reviewer 1 Report
This is a data-research paper that is addressing an interesting topic. The authors selected a reasonable plant to study, though its relationship to the fight against climate change is unclear.
The introduction is far too brief, requiring the readers to determine what kind of tree Casuarina glauca Sieb. ex Spreng might be and to speculate on its use against climate change.
The remaining sections are clear and well constructed.
The resolution of the figures was far too low to be easily read. Some figures were completely unreadable. I assume this will be corrected during final edits.
Author Response
Dear Reviewer,
Thank you for sending your comments.
We have introduced all the requested changes, as indicated below.
We hope that this version meets the quality criteria to be published in this special issue of Plants, but will be available to introduce more changes as need arises. Attached the modified version.
Sincerely,
Ana I. Ribeiro Barros
Corresponding author (On behalf of all co-authors)
Lisbon 26 October 2022
*********************************************************
Reviewer comments and author responses
This is a data-research paper that is addressing an interesting topic. The authors selected a reasonable plant to study, though its relationship to the fight against climate change is unclear.
Response: Thank you for your positive evaluation.
The introduction is far too brief, requiring the readers to determine what kind of tree Casuarina glauca Sieb. ex Spreng might be and to speculate on its use against climate change.
Response: We have now included additional information to clarify this aspect.
The remaining sections are clear and well-constructed.
Response: Thank you.
The resolution of the figures was far too low to be easily read. Some figures were completely unreadable. I assume this will be corrected during final edits.
Response: Presumably, this is related to the integration of the figures in the word document, and to the conversion into a pdf file. We will follow this issue with the editorial team to ensure that, if the paper is accepted, the original files will be used.

Reviewer 2 Report
In Ñ‚he article I reviewed, the authors analyzed the photosynthetic or gans transcriptome of C. glauca, one of the first stress targets, with the aim of studying the molecular mechanisms underlying stress tolerance. In my opinion, the research is systematized and well-organized. The experiments detailed and the data reported appear generally solid. Data analyzes are logical and well-presented. The results obtained are interesting and in line with the discussion.
In summary, the paper is well put together and contains a wealth of interesting new data, so it will be of interest to other researchers working in this field as well as readers of this journal.
I recommend accepting the manuscript in present form
Author Response
Dear Reviewer,
Thank you for accepting the MS.
Meanwhile we have introduced the minor changes suggested by the other two reviewers. Attached the modified version.
Sincerely,
Ana I. Ribeiro Barros
Corresponding author (On behalf of all co-authors)
Lisbon 26 October 2022

Reviewer 3 Report
I have only minor comments to this well-planned, well-done, and well-written manuscript.
1. All figures are of insufficient resolution in the pdf version.
2. The source of information on gene annotation should be more clearly indicated. Automatic annotation fails in quite many cases.
3. Line 21: together with
Author Response
Dear Reviewer,
Thank you for sending your comments.
We have introduced all the requested changes, as indicated below.
We hope that this version meets the quality criteria to be published in this special issue of Plants, but will be available to introduce more changes as need arises. Attached the modified version.
Sincerely,
Ana I. Ribeiro Barros
Corresponding author (On behalf of all co-authors)
Lisbon 26 October 2022
*********************************************************
Reviewer comments and author responses
I have only minor comments to this well-planned, well-done, and well-written manuscript.
- All figures are of insufficient resolution in the pdf version.
Response: As referred above, the insufficient resolution seems to be related to the integration of the figures in the word document, and to the conversion into a pdf file. We will follow this issue with the editorial team to ensure that, if the paper is accepted, the original files will be used.
- The source of information on gene annotation should be more clearly indicated. Automatic annotation fails in quite many cases.
Response: We have now clarified this aspect.
- Line 21: together with
Response: This has been changed accordingly.
